# Olfactory Learning in the Stingless Bee *Melipona eburnea* Friese (Apidae: Meliponini)

**DOI:** 10.3390/insects10110412

**Published:** 2019-11-18

**Authors:** Marisol Amaya-Márquez, Sergio Tusso, Juan Hernández, Juan Darío Jiménez, Harrington Wells, Charles I. Abramson

**Affiliations:** 1Instituto de Ciencias Naturales, Universidad Nacional de Colombia, Bogotá 111321, Colombia; juchernandezpe@unal.edu.co (J.H.); jimenezjuandario@gmail.com (J.D.J.); 2Science for Life Laboratories and Department of Evolutionary Biology, Norbyvägen 18D, Uppsala University, 75236 Uppsala, Sweden; situssog@gmail.com; 3Division of Evolutionary Biology, Faculty of Biology, LMU Munich, 82152 Grosshaderner Str. Planegg-Martinsried, Germany; 4Department of Biology, Tulsa University, Tulsa, OK 74104, USA; harrington-wells@utulsa.edu; 5Department of Psychology, Oklahoma State University, Stillwater, OK 74078, USA; charles.abramson@okstate.edu

**Keywords:** Free-moving-PER (fmPER), conditioning protocols, stingless-bees, learning, pollinators, scented-US, olfactory conditioning, cognitive ecology

## Abstract

Olfactory learning and floral scents are co-adaptive traits in the plant–pollinator relationship. However, how scent relates to cognition and learning in the diverse group of Neotropical stingless bees is largely unknown. Here we evaluated the ability of *Melipona eburnea* to be conditioned to scent using the proboscis extension reflex (PER) protocol. Stingless bees did not show PER while harnessed but were able to be PER conditioned to scent when free-to-move in a mini-cage (fmPER). We evaluated the effect of: 1) unconditioned stimulus (US) reward, and 2) previous scent–reward associations on olfactory learning performance. When using unscented-US, PER-responses were low on day 1, but using scented-US reward the olfactory PER-response increased on day 1. On day 2 PER performance greatly increased in bees that previously had experienced the same odor and reward combination, while bees that experienced a different odor on day 2 showed poor olfactory learning. Bees showed higher olfactory PER conditioning to guava than to mango odor. The effect of the unconditioned stimulus reward was not a significant factor in the model on day 2. This indicates that olfactory learning performance can increase via either taste receptors or accumulated experience with the same odor. Our results have application in agriculture and pollination ecology.

## 1. Introduction

Classical and operant conditioning are two fundamental aspects of learning and memory [1,2], which help to explain how organisms perceive, process, and react to environmental information [3]. Studies of classical conditioning in different types of organisms are relevant in comparative models of cognitive architectures for various taxa [4]. Similar to vertebrate studies, a very limited number of species largely drives our understanding of insect learning and memory. The fruit fly (*Drosophila melanogaster*), honey bee (*Apis mellifera*), and bumble bees (*Bombus* spp.) have become robust and influential insect model systems because they are amenable to highly controlled experimental manipulations [5,6], such as the proboscis extension reflex (PER), sting extension reflex (SER), artificial flower patch, shuttle box, and Y-maze techniques [7,8,9,10,11]. In bees, the proboscis extension reflex (PER), proposed first by [12] and refined by [13], has emerged as a powerful Pavlovian conditioning protocol, mainly in studies of olfactory conditioning of honey bees [5]. Its power lies in its usefulness for testing appetitive learning and associated memory at levels of analysis that range from behavioral to molecular [14,15,16,17,18]. The technique has also provided controlled conditions that have led to use of honey bees as an alcoholism model [19,20,21], and to test the basis for pesticide effects on the fundamentals of learning [22] with a wide range of potential applications in biomedical and agricultural research [23]. For example, pesticide exposure is correlated with changes in the antenna sensillae related to both calcium binding protein expression and free calcium levels in the brain [24], even when neurological functions such as flight and orientation in the environment are not noticeably impaired. Not only do pesticides affect use of conditioned stimulus (CS) odor, they also affect error rates associated with color CS reward associations. This appears to be due to both developmental issues involving the morphometrics of ommatidia facets and with the expression of the opsin and rhodopsin visual proteins in the ommatidia [11]. The European honey bee (*Apis mellifera* var. *ligustica*), as a biological model for olfactory learning, typically reaches 80% to 90% correct discrimination between odors by exhibiting the conditioned response (CR) of proboscis extension after just a few trials [1,25].

The use of PER with bumble bees is rare but has also been successfully employed in a few studies [26,27,28,29,30,31]. The earlier studies using *B. terrestris* showed at best a 30% proboscis response rate to the CS. This rate could be increased when the scent was added to apiary syrup, which nevertheless complicated the basic Pavlovian conditioning premise. However, more recent studies [26] report discrimination rates up to 60% when inter-trial-intervals (ITI) are elongated. Additionally, studies using other *Bombus* species show higher proboscis response rates in the 60% to 80% range [27,31]. Based on these PER studies, reported associative learning abilities by bumblebees appear to mirror those of honey bees.

The proboscis extension technique has been tried with the stingless bees of the family Apidae, but the results of those studies have not shown a consistent pattern of PER conditioning response. An early study with *Melipona scutellaris* [32] showed no learning using the PER paradigm, even though the same scientists demonstrated odor association in Africanized honey bees using the same methods [33]. The PER technique also failed to elicit differential conditioning using *Scaptotrigona aff. depilis* [34] and *Tetragonisca angustula* [35] as the test subjects. However, classical conditioning with *Melipona quadrifasciata* using the PER methodology was reported [34], but only with an average of 37% proboscis extension to the correct CS at the end of training. A study on olfactory learning using PER conditioning in two Paleotropical stingless bee species, *Meliponula ferruginea* and *M. bocandei*, reported 60% and 17% of conditioned response for these two species, respectively [36]. Furthermore, subsequent findings with *M*. *quadrifasciata* and *T. angustula* showed that these stingless bee species were highly influenced in food choice by odors previously experienced in the hive or in a foraging context [35,37]. Interestingly, when *T. angustula* bees experienced scented food in the hive prior to PER conditioning, the bees responded to discriminant olfactory conditioning [35]. Additionally, having the CS scent in the US improved the PER response rate of harnessed *M. quadrifasciata* [37].

Bees depend on pollen and nectar resources for larval development and adult metabolic functioning and thus are frequent floral visitors. Co-adaptive traits for the plant–pollinator functioning includes cognitive processes, such as associative learning and memory of flower cues, that help pollinator bees to discriminate and choose flowers, affecting behavior and pollination [38,39]. Floral scent and insect olfaction are crucial traits in the evolution of plant–pollinator relationships [40]. Bees learn to associate particular floral scents with specific floral rewards, returning to the most rewarding plants guided by odor memory. Therefore, understanding of associative learning in bees is fundamental in floral choice and floral constancy [15] and central to the development and management of new commercial pollinator species.

A recent study [41] showed that locally diverse bee pollinator assemblages achieved sustained crop pollination at larger landscape scales over time. In addition to conserving bee species richness for safeguarding pollination services, we also need to understand the diverse cognitive architectures of wild bee species. At present, most of what we know about olfactory learning has been widely studied in the honey bee, while the process is largely unknown for most bee species.

The stingless bees (Hymenoptera, Apidae, Meliponini) are diverse group of eusocial, corbiculate bees, with around 500 species distributed in the tropical and southern subtropical regions of the world [42], with potential to be used as commercial pollinators [43,44,45,46]. In the Neotropics, the stingless bee *Melipona eburnea* is an important flower pollinator of Andean plants [47]. It is similar to the honey bee (*Apis mellifera*) both in generalist foraging behavior of the colony, and also in the corporal size of forager bees. This species is widely distributed with records in Venezuela, Colombia, Ecuador, Peru, and Brazil [48].

Given the potential of *M. eburnea* to be used as a commercial pollinator, and the relevance of odor learning in understanding the foraging ecology of bees, we conducted a study on olfactory conditioning of this species. First, we asked if: (i) the stingless bees *M. eburnea* can be olfactory conditioned using the laboratory PER protocol. If so, we then wanted to know: (ii) the effect of using scented unconditioned stimulus (US) in the CS–US contingency on olfactory learning, and (iii) the effect of previously experienced association of scent (CS) and reward (US) on posterior performance in olfactory conditioning. We assessed the effect of reward and previous olfactory experience on olfactory learning in stingless bees. The conditioned stimulus (CS) and the unconditioned stimulus (US) are represented in nature by floral aroma (CS) and nectar (US). However, nectar is more than a sucrose solution and often contains volatiles that may affect the olfactory learning response of bees and quality of pollination. We investigated this effect using the PER conditioning paradigm. Workers of this species forage for several days and face frequent changes in the floral landscape, including changes in floral volatiles and aromas as different plant species bloom. Thus, we evaluated the effect of previous scent association on posterior olfactory PER performance.

## 2. Materials and Methods

In the first phase of the study we conducted a series of experiments to address whether the stingless bee *Melipona eburnea* was able to be olfactory conditioned using the laboratory PER protocol as it has been implemented for the honey bee *A. mellifera* [13].

### 2.1. Olfactory Conditioning of Harnessed Bees Using the Proboscis Extension Reflex (PER)

We evaluated absolute and differential conditioning using harnessed bees [32,34,37]. The terms used here follow previous definitions of conditioning [49]. We used the sugar response threshold (SRT) test [50] to determine the sugar concentration value of the US. The olfactory conditioning with harnessed stingless bees did not work, with no bees exhibiting the conditioned response (PER = 0 in all trials) both in absolute and in differential conditioning experiments (Appendix A), although the control using honey bees verified that the method was employed effectively (Appendix A).

### 2.2. Olfactory Conditioning of Free-Moving (fm) Bees Using the Proboscis Extension Reflex PER (fmPER)

Based on the negative results of the set of experiments with harnessed *M. eburnea* workers, we developed a PER method using a bee free-to-move within a mini-cage (fmPER) rather than a bee harnessed in a tube. The rationale behind the use of the cage was to remove the harnessed condition and thus potentially see conditioned PER response. Using the free-moving PER technique, bees responded positively exhibiting both the unconditioned response (UR) (i.e., extending the proboscis when the antenna was touched with a sucrose solution), and the conditioned response (CR) (i.e., extending the proboscis in response to a conditioned scent before receiving the sucrose solution reward (see the method below)).

### 2.3. Effect of Reward and Previous Scent–Reward Experience on Olfactory PER Conditioning

In the second phase of the study we conducted eight experiments of classical absolute scent conditioning [32,33] using a fmPER. The experiments aimed to evaluate: (1) the effect of using scented reward in CS–US contingency, and (2) the effect of previous bee experience with scent–reward associations on olfactory PER performance. The study took place in the Cordillera Oriental of the Colombian Andes, in the Department of Cundinamarca, municipality of Pandi at Guanani, from July through September 2015.

We collected *M. eburnea* workers from the hive entrance as they left for foraging using a wire framed mesh cylinder (10 cm in diameter, 25 cm long). The bees were placed in separate cages. Each cage was a glass vial (4 cm long, 2 cm ID) with a plastic screen on one end (Figure 1), which replaced the classic 9 mm ID tube used to harness bees in PER experiments [13,20]. Each bee was fed 5 μL 35% (w/v) sucrose solution and then, to reduce potential stress, placed in darkness at room temperature for 3 h prior to the experiment. We evaluated classical PER conditioning of *M. eburnea* to scent following the PER protocol used with the honey bee, *Apis mellifera* [13,33]. We used bees that: (1) did not show spontaneous response to the CS scents used in the experiments, and (2) did show the UR of extending their proboscis when the antenna was in contact with the sucrose solution.

As conditioned stimuli (CS) we used guava and mango scents, and as unconditioned stimuli (US) we used scented and unscented rewards of 50% (w/v) sucrose solutions. We used syringes to dispense the CS [5,51]; we attached to the syringe’s plunger a nylon mesh bag (1.5 cm side) containing the source of aroma. For guava we used a 1 cm cube of fruit mesocarp, and for mango we used 30 flowers that were easily packed in the bag. All experiments (1–8) examined conditioning over a two-day interval. With the aim to evaluate the effect of previous scent–reward associations on olfactory conditioning, we controlled for experience with the same or different odor. Half of the experiments (1–4) used a single scent so the bees experienced the same odor each day. The other half of the experiments (5–8) used two scents so the bees experienced a different odor each day (Table 1).

The CS–US stimuli contingency was presented with a non-overlapping design: 2 s of exposition to the CS scent followed by 3 s of US reward. The US was presented by first touching a subject’s antennae with the 50% w/v sucrose solution. Each day’s conditioning lasted about 1 h for each bee. The intertrial time intervals (ITI) lasted 10 min. We used 12 conditioning trials on day 1 and 6 trials on day 2. The second day’s conditioning occurred 24 h after the initiation of the first day’s conditioning period.

The conditioned response PER was marked as a binary: (1) for positive PER responses and (0) for non-responses; this PER response was used to quantify olfactory learning. The bee’s response to the US was also marked as a binary: (1) when the bee fed and (0) for non-feeding; this behavior evaluated the physiological condition of bees at each trial.

To assess whether bees were responding to any odor rather than only to the particular conditioned CS, we conducted an internal control test for each experiment. For experiments that used a single scent (Experiments 1–4) after the 6th conditioning trial on the second day (CS+), a single trial uncoupled with the US was performed using lavender as the novel CS (L−), and then a single trial uncoupled with the US was performed with the conditioned CS of the experiment (CS−) (mango for Experiments 1 and 3 and guava for Experiments 2 and 4). For experiments that used two scents (Experiments 5–8), after the 6th conditioning trial on the second day a single trial uncoupled with the US was performed using lavender as the novel CS (L−), then a single trial uncoupled with the US was performed with the CS of the second day (CS−), and finally a single trial uncoupled with the US was performed with the CS of the first day (CS−) (lavender, guava, and mango for Experiment 5; lavender, mango, and guava for Experiment 6; Lavender, mango, and guava for Experiment 7; lavender, guava, and mango for Experiment 8).

Statistical Analysis: The response to the US was analyzed via a repeated measures MANOVA with Experiment, Day, and Interaction (Exp. × Day) as factors [52]. The PER response for specific conditioned CS was analyzed with a t-test [53]. We compared the last trial of the second day of each experiment PER CS response paired with the US (CS+) against a novel CS (L−) and then the CS cue(s) unpaired with the US (CS−).

To evaluate the effect of reward and previous scent–reward associations on olfactory learning, we analyzed the conditioned response PER. We fitted a generalized linear mixed model (GLMM) with the conditioned PER as the response variable, and US reward, odor, previous scent–reward experience, and trial as fixed factors. We used individual bee as a random factor. We used a binomial distribution and the logit link function. Two separated GLMM models were run: (1) To evaluate the effect of the unconditioned stimulus (US) reward on olfactory PER conditioning, we used the conditioned PER response on day 1 for the six first trials (the bees responses declined after trial 6, possibly due to satiation); (2) to evaluate the effect of previous scent–reward associations, on posterior olfactory PER conditioning, we used the conditioned PER response on day 2. In both cases, information criteria in model selection was used. We used AIC (Akaike Information Criteria) and BIC (Bayes Information Criteria) to choose the final model, applying the elimination method, which starts with the most complex model including all factors and all interactions and ends with the most parsimonious and robust model. At each step we used chi square to evaluate if there were significant differences between the previous model and the last one, after elimination of the non-significant terms at a probability level of 0.05 [54]. We calculated the dispersion and the significance of the main factors and interactions in the most parsimonious model using the Chi square of Wald [55] and the significance among levels of each factor. The GLMM analyses were performed in R version 3.6.1 [56]. We used the functions “drop1” from the stats package, “summary” from the base R package [56], “glmer” from the lme4 package [57], “Anova” from the car package [58], and “effect” from the effects package [55].

## 3. Results

### Effect of Reward and Previous Scent–Reward Experience on Olfactory PER Conditioning

*Bees response to the US reward*: The bees’ response to the US in all eight experiments on both days was high (Figure 2). There was no significant effect of Experiment (F_7,64_ = 1.456, *p* = 0.1991), Day (F_1,64_ = 0.1054, *p* = 0.7465), or Interaction (F_7,64_ = 1.7954, *p* = 0.1035) effect. The average response rate for the US over all experiments (both days) was 91%.

*Conditioned response PER to reward and previous scent–reward associations*: Two different models were analyzed to investigate if reward (scented-US or unscented-US) and previous experience (same or different odor) were explicative factors of the conditioned response PER. The effect of training trials, type of odor, and bee were also included in the analyses.

**Model 1: Effect of the unconditioned stimulus (US) reward on olfactory PER conditioning**: On day 1, the stingless bees *M. eburnea* showed low conditioned olfactory PER response. This response increased when a scented sucrose reward was used as the unconditioned stimulus (US) (chi sq = 20.9994, df = 1, *p* < 0.0001 (Figure 3)). The increment occurred along the training trials (chi sq = 41.3564, df = 1, *p* < 0.0001), and the increment was even higher when the conditioned odor was guava (G) than when it was mango (M) (chi sq = 5.6365, df = 1, *p* < 0.05). (Model: Conditioned_Response ~ Trials + Reward + Odor + (1 | Bee_Total)), *p* < 0.0001; Appendix A). The significance of these factors by levels are shown in Table 2.

**Model 2: Effect of previous scent (CS)-reward (US) associations on olfactory PER conditioning**: On day 2 the stingless bees *M. eburnea* showed a higher PER response when they had previous experience with same odor (chi sq = 18.2407, df = 1, *p* < 0.0001 (Figure 4)) than when the previous experience was with a different odor. The increment occurred along the training trials (chi sq = 24.1110, df = 1, *p* < 0.0001). When the bees’ previous experience was with a different odor and the conditioned odor the second day was guava (G), bees learned (chi sq = 3.6954, df = 1, *p* < 0.05), while when mango was the odor used the second day, the bees did not show the conditioned PER response (Figure 4, Table 3; Model: Conditioned_Response ~ Trials + Experience + Odor + (1 | Bee_Total) + Trials: Experience), *p* < 0.0001; Appendix A). The significance of these factors by levels is shown in Table 3.

*Control tests for specific CS scents learning and memory:* The control tests given to the bees after the sixth conditioning trial on day 2 showed that conditioned response PER was specific to the conditioned scent CS.

For experiments 1 and 2, PER response was tested first against a novel CS– (lavender scent) and then the CS of the experiment (CS+) in both cases unpaired with the US (sucrose). The response rate to the unpaired CS– trial was significantly less than the second day average response to the paired CS+ trials (Experiment 1: t_8_ = 4.8077, *p* = 0.0013; Experiment 2: t_8_ = 3.2500, *p* = 0.0117). However, this was not the case for the unpaired CS+ trial; it did not differ significantly from the second day average response to the paired CS+ trials (Experiment 1: t_8_ = 1.6058, *p* = 0.1470; Experiment 2: t_8_ = 0.5067, *p* = 0.6260 (Appendix A)). These results indicate the bees’ response was specific to the scent conditioned and it was not a sensitization or generalization response.

For experiments 3 and 4, PER response was tested first against a novel CS– (lavender scent) and then the CS+ in both cases unpaired with the US. The response rate to the unpaired CS– trial was significantly less than the second day average response to the paired CS+ trials (Experiment 3: t_8_ = 2.8267, *p* = 0.0222; Experiment 4: t_8_ = 5.068, *p* = 0.0148). However, this was not the case for the unpaired CS+ trial; it did not differ significantly from the second day average response to the paired CS+ trials (Experiment 3: t_8_ = 0.2645, *p* = 0.7981; Experiment 4: t_8_ = 0.0002, *p* = 0.9988). Here, as in the previous two experiments, the results indicate the PER response of the bees was specific to the conditioned scent (mango in Experiment 3 and guava in Experiment 4) and was not a sensitization or generalization response. These results indicate that bees associated the CS+ with the reward. Further, in the experiments using the same scent each day (Experiments 1–4), it was clear that at the first trial of day 2, the bees started with memory for the CS+ of the previous day (Appendix A).

Like the first four experiments, immediately following the last trial on the second day of Experiments 5 and 6, PER response was tested first against a novel CS– (lavender scent), then against each of the CS+ scents separately, and in all three cases the CS was unpaired with the US. The response in Experiment 5 to the unpaired lavender CS was significantly different from both the CS+ paired-test with mango (Experiment 5: t_9_ = 8.6979, *p* < 0.0001) and with guava (Experiment 6: t_7_ = 7.3641, *p* = 0.0002). The response rate was not significantly different between either the mango CS+ paired and unpaired (Experiment 5: t_9_ = 2.1875, *p* = 0.0564) or guava CS+ paired and unpaired (Experiment 5: t_7_ = 0.5707, *p* = 0.5860). For the same set of tests in Experiment 6, each was significant (Experiment 6: in each case t_10_ > 3.08711, *p* < 0.01). Again, immediately following the last trial on the second day of Experiments 7 and 8, PER response was tested first against a novel CS– (lavender scent), then against each of the CS+ scents separately, and in all three cases the CS was unpaired with the US (Appendix A).

The response in Experiment 7 to the unpaired lavender CS– was significantly different from both the CS+ paired-test with mango on day 1 (Experiment 7: t_10_ = 2.3770, *p* = 0.0388) and with guava on day 2 (Experiment 7: t_10_ = 3.0126, *p* = 0.0131). The PER response was not significantly different between either the mango CS+ paired and unpaired (Experiment 7: t_9_ = 0.0984, *p* = 0.9239) or guava CS+ paired and unpaired (Experiment 7: t_7_ = 1.7468, *p* = 0.1107). Results in Experiment 8 were like those of Experiment 7. The response in Experiment 8 to the unpaired lavender CS– was significantly different from both the CS+ paired-test with mango (Experiment 8: t_10_ = 2.3000, *p* = 0.0443) and with guava (Experiment 8: t_11_ = 5.0931, *p* = 0.0003). The PER response was not significantly different between either the mango CS+ paired and unpaired (Experiment 8: t_10_ = 1.3880, *p* = 0.1953) or guava CS+ paired and unpaired (Experiment 8: t_11_ = 0. 5821, *p* = 0.5722). These results indicate that the PER responses of the bees were specific to both the odor experienced on day 1 (mango) or on day 2 (guava) for Experiment 7, and on day 1 (guava) and day 2 (mango) for Experiment 8 and did not correspond to either sensitization or generalization responses. These results indicate that the stingless bees *M. eburnea* is able to retain long term memory for odors. Contrary to what was observed in the experiments using the same scent each day (Experiments 1–4), here in the experiments using a different scent each day (Experiments 5–8), the bees conditioned response PER started at zero in the first trial of day 2 (Appendix A).

## 4. Discussion

Similar to honey bees and bumble bees, but unlike solitary bee species [59], we show that the stingless bee *M. eburnea* presents a PER to US sugar solutions. Using the PER, we evaluated the sucrose threshold response (STR) of the stingless bees; 67% of the bees showed a STR above 30% w/v sucrose solution concentration. However, a wide variation in response was recorded (1%–50% w/v). In the honey bee, *A. mellifera*, it has been reported that differences in STR are correlated with a forager role [50], which may account for some of the variation we observed in *M. eburnea*.

When we used the traditional harnessed protocol for PER conditioning [13], *M. eburnea* did not show olfactory learning, either with absolute or differential conditioning. In contrast, our control tests using the Africanized honey bee (*A. mellifera* var. *scutellata*) were similar to those reported for this species in other studies [33,36]. The lack of response led us to try a modified PER protocol that used mini-cages in which the bees had free-movement instead of using harnessed bees in tubes.

We found that the response to the US of the stingless bees *M. eburnea* freely moving in a mini-cage was high in all eight experiments, ranging from about 85% to nearly 100%. Bees consumed the nectar that was offered to them in each trial. Based on these results, we ruled out that any lack of response was due to low motivation or to a poor physiological condition of the tested bees. The results are comparable to that reported for honey bees in harnessed PER experiments [60]. Thus, any differences observed between CS responses of *M. eburnea* reported here and *A. mellifera* reported in the literature cannot be simply ascribed to the experimental apparatus.

In this study we found olfactory PER conditioning of the stingless bees *M. eburnea* depended on the protocol used. This finding can explain inconsistences among studies reporting differences in the capacity of olfactory conditioning using PER across stingless bee species [32,34,36]. Our result supports the skepticism previously expressed by other authors on the adequacy of the protocol standardized for the honey bee, *A. mellifera*, when testing stingless bee species [32,35,36]. Thus, the lack of olfactory PER conditioning in some of the Neotropical stingless bees may be the result of the apparatus to test PER conditioning, rather than a cognitive trait of the tested bee species. Here we presented a novel apparatus to test classical olfactory learning in the stingless bees (fmPER), which contributes to widen the array of methods [61,62] available for the study of cognitive architectures in different groups of bees.

On day 1, the experiments examined the effect of the unconditioned stimulus (US) reward on olfactory PER performance. We found that the probability of the conditioned PER was affected by the US reward. The PER response was low when the bees were trained with the procedure for honey bees, where the CS is a scent cue preceding the unscented sucrose US. The rapid learning on day one reported for honey bees did not occur. Nevertheless, *M. eburnea* clearly associated the CS with the subsequent US indicated by the control tests for specific CS+ response, and from the initial trial on day 2 for bees that were trained with the same scent, indicating they developed long term memory for the CS+ experienced on day 1. However, when the CS scent was added to the US there was a notable rise of olfactory PER response. The increment occurred along the training trials indicating a learning process. By the sixth trial, stingless bees reached an average probability to elicit the conditioned PER to 75% when the conditioned odor was guava, while for mango the bees reached at the same trial an average probability of 40% to learn the odor. The increment in learning by using scented-US was differential to odor, been higher for guava than for mango.

For colonial species, CS–US associations should take place at both the flower [63] and at the colony [64,65,66,67]. Indeed, having experienced a scent in the hive or in the nectar reward has been shown to significantly improve harnessed PER response in some stingless bees [35,37]; furthermore, bumble bees fed with scented sucrose before olfactory conditioning exhibited a better PER performance than those who received unscented sucrose [31]. Our results are coherent with the findings in [35,37,68], indicating that taste receptors may play a larger role in olfactory learning in bees, and that type of odor matters when the CS is in the US. Integration of gustatory and olfactory pathways facilitates rapid learning of food cues [38,69]. In the context of pollination, flower aroma represents the CS and nectar represents the US. In classical absolute olfactory conditioning, a CS scent is paired with unscented US sucrose solution reward; this protocol translated to a naturalistic situation assumes nectar is unscented and flower aroma emanates from other floral structures (e.g., osmophores, petals, sepals). However, nectar is more than a sucrose solution and often contains volatiles that affect pollinator olfactory response and behavior, and thus the quality of pollination [70,71]. Nectar volatiles play a role in attraction and deterrence of pollinators [72]. Further, volatiles produced by microorganisms present in the nectar affect attraction and visitation patterns of pollinators [73,74,75]. However, it is not clear if the behavior of pollinators that are attracted or repelled to flowers is caused by the taste and/or scent of the compounds in the nectar [70,71]. Our study showed that scented-US increased olfactory learning and increased PER response was differential to odor type, indicating that stingless bees *M. eburnea* respond to the compounds present in the sucrose solution differentially.

On day 2, the experiments examined whether bees were simply responding to any CS odor on day 2, rather than to the CS odor experienced on day 1. The probability of the conditioned PER response was greatly affected by experience; those bees that had experienced the same odor the previous day exhibited a higher performance than bees that had experienced a different odor. Interestingly, in the group of bees that experienced a different odor each day, when guava was the odor used in the conditioning procedure on day 2, the bees along training trials adapted to the new situation, and by the sixth trial 40% bees had learned the odor. When mango was the odor conditioned on day 2, the bees did not learn the mango odor. We hypothesize that these results may be influenced by the floral resources used by *M. eburnea.* Previous studies on the floral resources used by *M. eburnea* show that guava is an important resource for this stingless bee species year round, while evidence of mango use was not recorded [47]. Interestingly, guava (*Psidium guajava*) is a Neotropical species as well as the stingless bee *M. eburnea*, while mango (*Mangifera indica*) is a Paleotropical species. Differences in learning responses may be sensory biased, possibly due to evolved innate co-adaptive differences when exploiting food [38,76,77]. Future research on the biological significance of cues used in olfactory conditioning may contribute better understanding on bees’ adaptation to environmental change.

The bees that were trained to a different scent presented no conditioned response on the first trial of day 2. The results show that these bees can learn some new CS–US associations, and that they are not simply responding to any CS in the artificial environment. Our results suggest olfactory associations with a reward are a cognitive investment beneficial when the floral resource is available in the floral landscape for a prolonged period of time but can be a cognitive cost if the floral landscape changes [78,79]. Flowers emit aromas that help bees find resources [39,40,63,80], producing mixtures of nectar volatiles unique for each plant species [80,81]. Stingless bee foragers exhibit floral constancy for several days [82] even though they are generalists as a colony. Pollen resources present in pollen-pots and honey-pots from nests of *M. eburnea* show the majority of plant resources used correspond to trees rather than to herbaceous plants [47]. Trees bloom producing abundant flowers that remain in the landscape for a long time, such as the “cornucopia” flowering pattern of Bignoniaceae trees [83]. Under this ecological scenario, the use of previously acquired information on floral scent is useful in foraging efficiency.

The individual bee was a factor affecting the conditioned PER response in both models. This indicates that individual bees differed in olfactory learning performance in ways that are not explained by the fixed factors. This “bee personality” may be the result of an extensive number of factors ranging from genetics to age and health condition. Here we did not control for those variables; however, we only used bees that were foraging, activity that can be taken as a proxy of bee age since this foraging task in Meliponini bees, like in the honey bee, is performed only in the last phase of the life cycle.

Work with the stingless bees using odor CS cues is suggestive that there may be some fundamental behavioral differences from our traditional *Apis* and *Bombus* insect bee model systems [32,34,35,36,37]. Our study also suggests this is true. In addition, a study by [76] found that innate color preference of stingless bees may differ from *Apis* and *Bombus*. Color preference of naïve *Apis cerana* and *Bombus terrestris* is to the shorter-wavelength color stimuli. In contrast, naïve *Tetragonula iridipennis* favor violet and blue-green stimuli, but not shorter or longer wavelengths in general. This may be due to either evolved innate differences when processing visual stimuli, or evolutionary differences in visual receptor spectral sensitivity. Regardless, reward correlated learning of color CS readily overrides innate color preferences [76]. Our study is contributing information on olfactory learning of the stingless bee *M. eburnea*, which is an important pollinator in the region, and according to the IUCN list, the conservation status of the species is vulnerable due to human hunting [84].

## 5. Conclusions

Classical conditioning of the proboscis extension reflex (PER) in the stingless bee *Melipona eburnea* depended on the protocol employed. Specifically, the conditioned PER was not observed in the experiments using harnessed bees but was observed in experiments conducted with free-moving bees (free-moving PER protocol: fmPER). This result explains the inconsistences reported in the literature on the ability of stingless bees to conditioned PER to scent. The new protocol presented here may be useful for studying olfactory learning in other species of stingless bees.

The use of the CS in the US (i.e., scented-US) showed that in the stingless bee *M. eburnea*, taste receptors play a role in olfactory learning. When using scented-US in the CS–US pairing, *M. eburnea* bees increased the conditioned PER response since the first trials on day 1. When using unscented-US, the stingless bees’ response was low on the first trials, but the PER response increased on day 2 trials. Interestingly, on the day 2 model, the effect of reward was not significant, indicating that olfactory learning can improve either via taste receptors or previous experience with the same odor. The use of scented-US allowed the stingless bees to learn a different scent each day.

The olfactory acquisition pattern recorded in this study with *M. eburnea* when the bees were trained with the procedure for honey bees presenting the CS scent before the unscented sucrose US differs from the honey bee (*Apis mellifera*), in which olfactory learning using PER is detectable from the first trials. Future research evaluating this difference in olfactory learning between the two eusocial species is warranted. Our results are of interest in the understanding of the cognitive ecology of bees and have application in agriculture and in pollination ecology.

## Figures and Tables

**Figure 1 insects-10-00412-f001:**
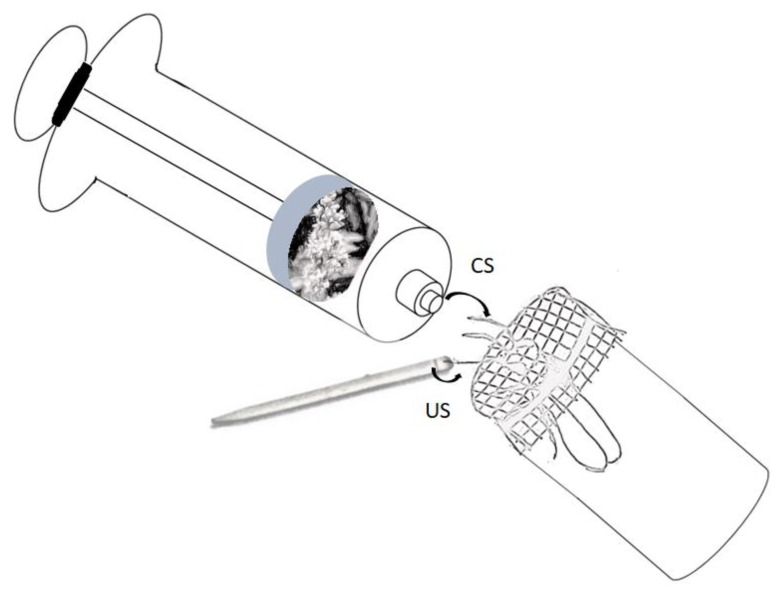
Apparatus used in scent conditioning in the stingless bee *Melipona eburnea* using the proboscis extension reflex (PER) with free-moving bees (fmPER). A bee housed in a glass vial in which the cap is replaced with plastic mesh. The bee is exposed to the trained scent (conditioned stimulus (CS)), and a plume of scented airflow is dispensed with a syringe. The unconditioned response is elicited by touching the antenna with the unconditioned stimulus (US). Conditioned PER occurs when the bee extends its proboscis through the mesh in response to the CS presented before receiving the US. The US was provided with the tip of a toothpick.

**Figure 2 insects-10-00412-f002:**
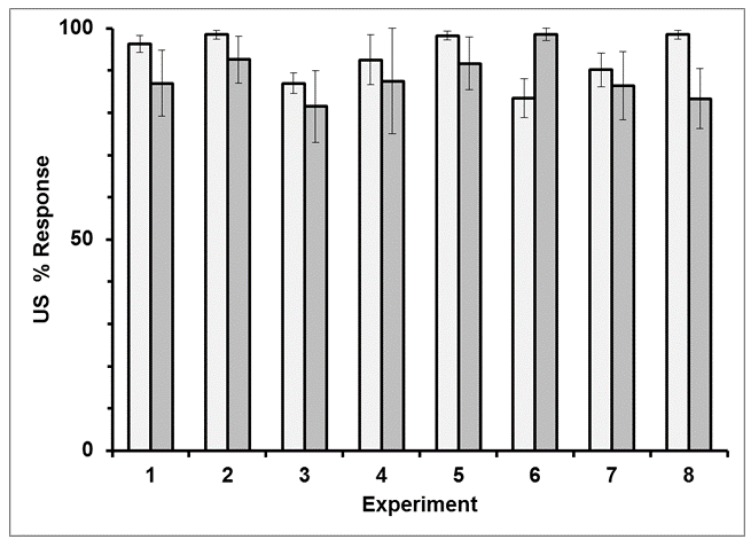
Average US PER response (±SE) in all eight experiments for day 1 (light bars) and day 2 (dark bars) using caged stingless bees, *Melipona eburnea.*

**Figure 3 insects-10-00412-f003:**
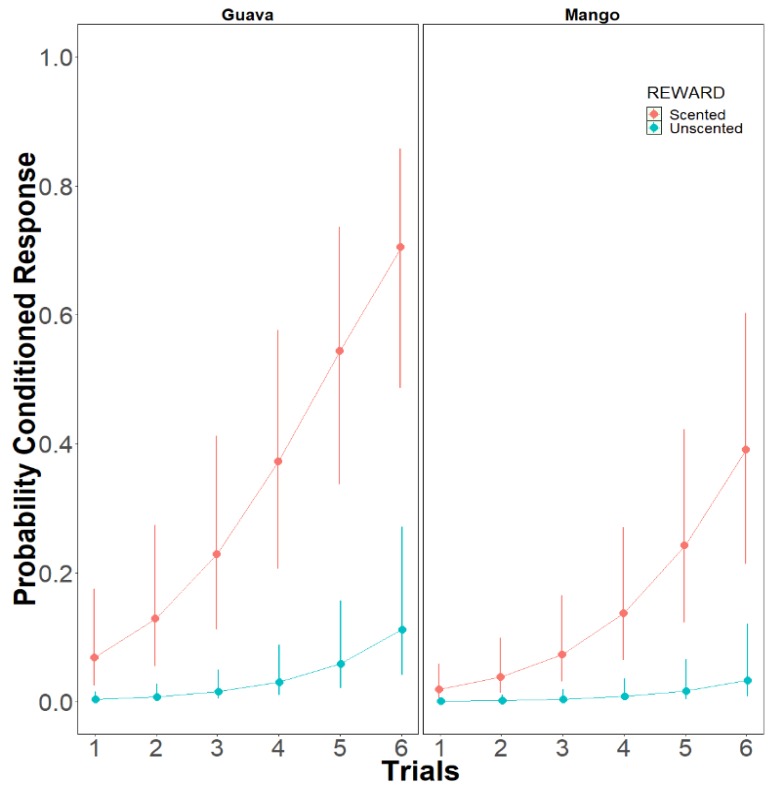
Probability of the conditioned response PER in the stingless bees, *Melipona eburnea* on day 1. Bees conditioned with scented-US reward increased the conditioned response. The increment occurred along the trials, being higher when the conditioned odor was guava (G) than when it was mango (M) (model: Conditioned Response ~ Trials + Reward + Odor + (1 | Bee_Total)). The 95% confidence intervals are shown.

**Figure 4 insects-10-00412-f004:**
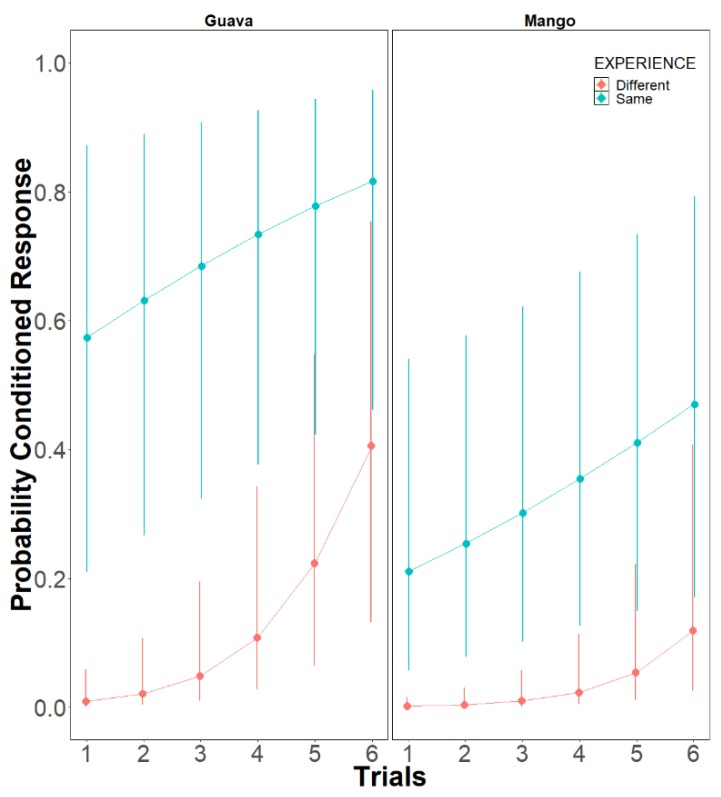
Probability of the conditioned response PER in the stingless bees, *Melipona eburnean*, on day 2. Bees that experienced the same odor the day before increased the conditioned response. The increment occurred along the trials, being higher when the conditioned odor was guava (G) than when it was mango (M) (model: Conditioned Response ~ Trials + Experience + Odor + (1 | Bee_Total) + Trials: Experience). The 95% confidence intervals are shown.

**Table 1 insects-10-00412-t001:** Experiments of classical absolute conditioning conducted on two consecutive days to assess the effect of scented-US reward and previous olfactory experience on olfactory learning in the stingless bee *Melipona eburnea*. The conditioned stimuli (CS) were mango (M) and guava (G); the unconditioned stimuli (US) were unscented-US (U) and scented-US (S). Experience with previous scent–reward associations were recorded as: same (when a single odor was used each day) and different (when two scents were used, a different one each day). Experiment code letters refer to scent used day 1, scent used day 2, and type of reward: mango mango unscented (MMU), guava guava unscented (GGU), mango mango scented (MMS), guava guava scented (GGS), mango guava unscented (MGU), guava mango unscented (GMU), guava mango scented (GMS), and mango guava scented (MGS).

Experiment Number	Experiment Code	Bee Sample (n Size)		Conditioned Stimulus (CS) SCENT		Unconditioned Stimulus (US) REWARD		Previous Scent–Reward Experience
		Day 1	Day 2	Day 1	Day 2	Day 1	Day 2	
1	MMU	9	9	M	M	U	U	Same
2	GGU	11	9	G	G	U	U	Same
3	MMS	9	9	M	M	S	S	Same
4	GGS	10	4	G	G	S	S	Same
5	MGU	10	8	M	G	U	U	Different
6	GMU	11	11	G	M	U	U	Different
7	GMS	12	11	G	M	S	S	Different
8	MGS	11	11	M	G	S	S	Different

**Table 2 insects-10-00412-t002:** Comparative significance by levels of the main factors in the fitted model 1 (generalized linear mixed model (GLMM)), explaining the olfactory conditioned response PER in day 1 of the stingless bee *Melipona eburnea.*

Parameter	Estimate	Standard Error	*p*-Value
(Intercept)	−3.2982	0.6096	<0.0001
Trials	0.6948	0.1080	<0.0001
Reward_Unscented	−2.9399	0.6415	<0.0001
Odor	−1.3128	0.5529	<0.05

**Table 3 insects-10-00412-t003:** Comparative significance by levels of the main factors in the fitted model 2 (GLMM), explaining the olfactory conditioned response PER in day 2 of the stingless bee *Melipona eburnea.* ns: not significant.

Parameter	Estimate	Standard Error	*p*-Value
(Intercept)	−5.5531	1.1041	<0.0001
Trials	0.8618	0.1755	<0.0001
Experience_Same	5.6107	1.3137	<0.0001
Odor_Mango	−1.6141	0.8397	ns
Trials:Experience_Same	−0.6227	0.2128	<0.01

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
