# Peer review of "Olfactory Learning in the Stingless Bee Melipona eburnea Friese (Apidae: Meliponini)"

_insects, 2019, doi:10.3390/insects10110412_

Round 1
Reviewer 1 Report
The authors have presented a very interesting study on the olfactory learning paradigms in the stingless bees. The study is well designed and it is certainly important to understand the behavioral perspectives of all bee species to enhance our understanding of pollination. My minor suggestions are as follows –
1. Any particular reason why different scented oils were used for the various absolute and differential conditioning experiments?
2. Some important papers are missing in the general introduction such as:
Chakrabarti et al. 2015 Scientific Reports Field populations of native Indian honey bees from pesticide intensive agricultural landscape show signs of impaired olfaction.
Balamurali et al. 2018 A comparative analysis of colour preferences in temperate and tropical social bees
Palottini et al. 2018 Odor Learning and Its Experience-Dependent Modulation in the South American Native Bumblebee Bombus atratus (Hymenoptera: Apidae)
Chakrabarti et al. 2019 Chemosphere Pesticide induced visual abnormalities in Asian honey bees (Apis cerana L.) in intensive agricultural landscapes
3. The authors may choose to remove some of the figures (for example 3 and 4 etc.) where there was no response recorded. Instead using a flow diagram and/or table to simplify the methods and results will certainly help. It is quite confusing and difficult to follow the text, as there are so many experiments. A simple flow chart will be extremely helpful for both methods and results, in place of figures 3 and 4.
4. Please check the manuscript for typos – for example spelling of lavender in figure 9.
Author Response
Any particular reason why different scented oils were used for the various absolute and differential conditioning experiments?
When we undertook this project there was very limited understanding in the literature about the olfactory biology of this bee species. We tried a variety of scents to see which produced different levels of conditioning, as measured by the learning curves. However, because the harnessed bee protocol resulted in negative conditioning (both absolute and differential), we did not compare the stingless bees to different scents. We acknowledge this is an important point to be considered in future studies, now that we have a protocol (fmPER) to assess olfactory conditioning in the stingless bees. Thanks for the question.
Some important papers are missing in the general introduction such as: Chakrabarti et al. 2015 Scientific Reports Field populations of native Indian honey bees from pesticide intensive agricultural landscape show signs of impaired olfaction. Balamurali et al. 2018 A comparative analysis of colour preferences in temperate and tropical social bees Palottini et al. 2018 Odor Learning and Its Experience-Dependent Modulation in the -South American Native Bumblebee Bombus atratus(Hymenoptera: Apidae) Chakrabarti et al. 2019 Chemosphere Pesticide induced visual abnormalities in Asian honey bees (Apis cerana L.) in intensive agricultural landscapes.
Thank you for the suggestion. We consulted the papers and we found they enriched our paper. We added Balamurali et al. 2015. Now they are cited in the introduction and discussion [see Lines within brackets]
Chakrabarti et al. 2015 Scientific Reports Field populations of native Indian honey bees from pesticide intensive agricultural landscape show signs of impaired olfaction. [Line 53] Balamurali et al. 2018 A comparative analysis of colour preferences in temperate and tropical social bees [Lines 436, 442] Balamurali et al. 2015. Senses and signals: Evolution of floral signals, pollinator sensory systems and the structure of plant-pollinator interactions [Lines 87, 387, 412] Palottini et al. 2018 Odor Learning and Its Experience-Dependent Modulation in the -South American Native Bumblebee Bombus atratus(Hymenoptera: Apidae) [Lines 62, 66, 384] Chakrabarti et al. 2019 Chemosphere Pesticide induced visual abnormalities in Asian honey bees (Apis cerana L.) in intensive agricultural landscapes. [Line 57]The authors may choose to remove some of the figures (for example 3 and 4 etc.) where there was no response recorded. Instead using a flow diagram and/or table to simplify the methods and results will certainly help. It is quite confusing and difficult to follow the text, as there are so many experiments. A simple flow chart will be extremely helpful for both methods and results, in place of figures 3 and 4.
Thank you for this helpful suggestion. We removed all the figures associated with Olfactory conditioning of harnessed bees using the proboscis extension reflex (PER), that includes: Figures 3 and 4 where there was no response, Figure 5 Experimental positive control in differential conditioning in the Africanized honey bee, and Figure 6 on the sucrose threshold response. These results as well as the methods are now presented in Supplementary material 1. We now present a table with all the experiments (Table 1) to simplify the methods and increase clarity in the results presentation.
Please check the manuscript for typos – for example spelling of lavender in figure 9.
We reviewed the entire MS for typos and corrected them.
Thank you so much for time and effort. The paper has improved thanks to your and the other reviewer suggestions.

Reviewer 2 Report
This is an interesting study of conditional learning in a stingless bee, which may have implications for our understanding of the evolution of learning in bees and the development of commercial pollinators. However, the presentation of the study needs significant work. My main concern is with the statistical analyses. It is not clear to me what exactly the authors did. This is critical to interpreting the results and the quality the study. Other aspects of the writing are necessary to improve the clarity of the manuscript but less concerning. My key recommendations for improvement of the manuscript are detailed below.
Improve the rational for the study. Explain why conditional learning is so important to pollinators and the cultivation of a commercial pollinator more thoroughly. Right now, this rationale is only 1 sentence long. Then, explain the need for more diverse commercial pollinator assemblages. Finally, touch on the bias in our understanding of insect learning. This “narrative” should be mirrored in the discussion. Indeed, the rationale isn’t well explained in either the introduction or the discussion. Remove the unsuccessful tethered experiment. I do not feel that it adds to the manuscript, and in fact, it detracts from the main findings of the article. If you feel it is required to justify the experimental design used in the second part of the paper, then explain it as such. However, this could be done in one sentence rather than with a subsection in the methods and results, including 3 figures. Simply state that it did not work, with no bees exhibiting a response. (PER = 0 in all trials). If, however, the point of the paper was to test the methodology (as implied in paragraph 4 of the discussion, lines410-419), then that needs to be clear from the very beginning. There was no rationale given for the scents used. Some of the scents used (i.e., cinnamon and guava fruit) may not stimulate the chemoreceptors of these insects, since they have not historically been signals to which these bees may have evolved. An alternative explanation for the varied responses in experiments 5 and 6 (discussion, lines 442-449) may be explained by the use of a flower scent (mango) versus a fruit scent (guava). Could the tethering experiments failed because the bees do not detect those odors? Is there any prior evidence that these bees can detect these odors? The statistical analyses are not described in enough detail to determine if the data were correctly analyzed. The authors seem to confuse linear regression with ANOVA at one point (line 219), and from my reading of the text, regressions were not actually used to test for learning. Perhaps, you used an analysis of covariance or multiple regression with day as a factor and learning as a continuous variable? The analyses need to be described in much more detail before an assessment of their appropriateness can be made. Based on my understanding of the statistical analyses, the figures do not reflect how the data were analyzed. The figures should reflect the analyses, illustrating what factors were determined to be significantly different and how. I recommend that the authors use a different approach to illustrating their data.
Additional notes:
Line 103: explain why the bees were placed in darkness at room temperature before the test commenced.
Line 106: what do you mean by “basic”? Do you mean to say that it is required for studies like this? Explain why that would be.
Line 114: Odd word arrangement. I would recommend “…its proboscis completely.”
Line 121: Provide a rational e for the use of 50% (w/v) sucrose solution, even if it is explained in greater detail after you describe the results of your SRT
Line 129: Even if your methods have been previously described, a brief description of them is still required for the new audience.
Line 150: replace “works” with “workers”
Line 268: Reference where the MANOVA results are presented. The actual statistical parameters should be presented somewhere in the text.
Author Response
REVIEWER 2.
This is an interesting study of conditional learning in a stingless bee, which may have implications for our understanding of the evolution of learning in bees and the development of commercial pollinators. However, the presentation of the study needs significant work. My main concern is with the statistical analyses. It is not clear to me what exactly the authors did. This is critical to interpreting the results and the quality the study. Other aspects of the writing are necessary to improve the clarity of the manuscript but less concerning. My key recommendations for improvement of the manuscript are detailed below.
Improve the rational for the study. Explain why conditional learning is so important to pollinators and the cultivation of a commercial pollinator more thoroughly [1]. Right now, this rationale is only 1 sentence long. Then, explain the need for more diverse commercial pollinator assemblages [2]. Finally, touch on the bias in our understanding of insect learning [3]. This “narrative” should be mirrored in the discussion. Indeed, the rationale isn’t well explained in either the introduction or the discussion. Remove the unsuccessful tethered experiment. I do not feel that it adds to the manuscript, and in fact, it detracts from the main findings of the article. If you feel it is required to justify the experimental design used in the second part of the paper, then explain it as such. However, this could be done in one sentence rather than with a subsection in the methods and results, including 3 figures. Simply state that it did not work, with no bees exhibiting a response. (PER = 0 in all trials). If, however, the point of the paper was to test the methodology (as implied in paragraph 4 of the discussion, lines410-419), then that needs to be clear from the very beginning [4]. There was no rationale given for the scents used. Some of the scents used (i.e., cinnamon and guava fruit) may not stimulate the chem oreceptors of these insects, since they have not historically been signals to which these bees may have evolved. An alternative explanation for the varied responses in experiments 5 and 6 (discussion, lines 442-449) may be explained by the use of a flower scent (mango) versus a fruit scent (guava). Could the tethering experiments failed because the bees do not detect those odors? Is there any prior evidence that these bees can detect these odors? [5]. The statistical analyses are not described in enough detail to determine if the data were correctly analyzed. The authors seem to confuse linear regression with ANOVA at one point (line 219), and from my reading of the text, regressions were not actually used to test for learning. Perhaps, you used an analysis of covariance or multiple regression with day as a factor and learning as a continuous variable? The analyses need to be described in much more detail before an assessment of their appropriateness can be made. Based on my understanding of the statistical analyses, the figures do not reflect how the data were analyzed. The figures should reflect the analyses, illustrating what factors were determined to be significantly different and how. I recommend that the authors use a different approach to illustrating their data [6].
[1]. Explain why conditional learning is so important to pollinators and the cultivation of a commercial pollinator more thoroughly.
In the revised manuscript we now include a paragraph in the introduction explaining why conditional learning is so important for using bees in commercial pollination [Lines 84-92]
“Bees depend on pollen and nectar resources for larval development and adult metabolic functioning and thus are frequent floral visitors. Co-adaptive traits for the plant-pollinator functioning includes cognitive processes, such as associative learning and memory of flower cues, that help pollinator bees to discriminate and choose flowers, affecting behavior and pollination [38, 39]. Floral scent and insect olfaction are crucial traits in the evolution of plant-pollinator relationships [40]. Bees learn to associate particular floral scents with specific floral rewards, returning to the most rewarding plants guided by odor memory. Therefore, understanding of associative learning in bees is fundamental in floral choice and floral constancy [15] and central in the development and management of new commercial pollinator species.”
Then, explain the need for more diverse commercial pollinator assemblagesWe now include a paragraph in the introduction explaining the importance of count with diverse bee pollinator assemblages which connects with previous paragraph [Lines 93-97]
“A recent study [41] showed that locally diverse bee pollinator assemblages achieve sustain crop pollination at larger landscape scales over time. In addition to conserving bee species richness for safeguarding pollination services, we also need to understand the diverse cognitive architectures of wild bee species. At the present, most of what we know about olfactory learning has been widely studied in the honey bee, while the process is largely unknown for most bee species.”
Finally, touch on the bias in our understanding of insect learningThe two last lines of previous paragraph make reference to the bias in studies conducted with the honey bee insect model [Lines 96-97]. But also, in lines [ 39-44] it was mentioned that most of what with know about insect cognition has been conducted with a handful of species.
“Similar to vertebrate studies, a very limited number of species largely drives our understanding of insect learning and memory. The fruit fly (Drosophila melanogaster), honey bee (Apis mellifera), and bumble bees (Bombus spp.) have become robust and influential insect model systems because they are amenable to highly controlled experimental manipulations [5-6], such as the proboscis extension reflex (PER), sting extension reflex (SER), artificial flower patch, shuttle box, and Y-maze techniques [7-10, 11].”
Remove the unsuccessful tethered experiment. I do not feel that it adds to the manuscript, and in fact, it detracts from the main findings of the article. If you feel it is required to justify the experimental design used in the second part of the paper, then explain it as such. However, this could be done in one sentence rather than with a subsection in the methods and results, including 3 figures. Simply state that it did not work, with no bees exhibiting a response. (PER = 0 in all trials). If, however, the point of the paper was to test the methodology (as implied in paragraph 4 of the discussion, lines410-419), then that needs to be clear from the very beginning.Thank you for this insightful suggestion. We agree and now have one short paragraph in the main document [Lines 124-128]. Additional information with the negative results as well as the methods used are now presented in Supplementary material 1.
“We evaluated absolute and differential conditioning using harnessed bees [32, 34, 35]. The terms used here, follow previous definitions of conditioning [49]. The olfactory conditioning with harnessed stingless bees did not work, with no bees exhibiting the conditioned response (PER=0 in all trials) both in absolute and in differential conditioning experiments (Supp. 1 Figs. S1-S3), though the control using honey bees verified that the method was employed effectively (Suppl. 1 Figs. S4).“
There was no rationale given for the scents used. Some of the scents used (i.e., cinnamon and guava fruit) may not stimulate the chemoreceptors of these insects, since they have not historically been signals to which these bees may have evolved. An alternative explanation for the varied responses in experiments 5 and 6 (discussion, lines 442-449) may be explained by the use of a flower scent (mango) versus a fruit scent (guava). Could the tethering experiments failed because the bees do not detect those odors? Is there any prior evidence that these bees can detect these odors?At the start of this study nothing was known about odor preferences of the stingless bee Melipona eburnea. Based on the literature on olfactory learning in honey bees, they can learn any odor; we started with this assumption and tried a variety of scents with M. eburnea expecting different levels of conditioning, as measured by the learning curves. Respect to guava now with the new statistical analyses it was clear the stingless bee M. eburnea showed a higher olfactory performance for guava scent than for mango. Additionally, in the experiments in which a different odor was used each day, when mango was the new odor, bees did not show learning, while when guava was the new odor on day 2, the bees by the sixth trial have learned it. In the original manuscript we did not know how to explain these differential responses. Now, we have a hypothesis posed in two premises: 1) If the enhanced olfactory performance for guava comes from co-adaptation, 2) does the sensorial bias for certain resources help bees to cope with changes in floral landscape aromas?. We now discuss those findings: Guava and the stingless bee M. eburnea are Neotropical species, and the bee uses guava pollen, while evidence of mango pollen use was not found; mango is a Paleotropical species. The information on pollen use by M. eburnea was obtained from a previous study by Obregon & Nates 2014.
[Obregon, D.; Nates-Parra, G. Floral Preference of Melipona eburnea Friese (Hymenoptera: Apidae) in a Colombian Andean Region. Neotrop. Entomol. 2014, 43, 53–60. https://doi.org/10.1007/s13744-013-0172-y]
This subject is now discussed in the manuscript [Lines 399-414]:
“On day 2, the experiments examined whether bees were simply responding to any CS odor on day 2, rather than to the CS odor experienced on day 1. The probability of the conditioned PER response was greatly affected by experience, those bees that had experienced the same odor the previous day exhibited a higher performance than bees that had experienced a different odor. Interestingly, in the group of bees that experienced a different odor each day, when guava was the odor used in the conditioning procedure on day 2, the bees along training trials adapted to the new situation, by the sixth trial 40% bees had learned the odor. When mango was the odor conditioned on day 2, the bees did not learn mango odor. We hypothesize that these results may be influenced by the floral resources used by M. eburnea. Previous studies on the floral resources used by M. eburnea show that guava is an important resource for this stingless bee species year round, while evidence of mango use was not recorded [47]. Interestingly, guava (Psidium guajava) is a Neotropical species as well as the stingless bee M. eburnea, while mango (Mangifera indica) is a Paleotropical species. Differences in learning responses may be sensory biased possibly due to evolved innate co-adaptive differences when exploiting food [38, 83]. Future research on the biological significance of cues used in olfactory conditioning may contribute better understanding on bees’ adaptation to environmental change.”
The question on the effect of using flower or fruit’s scents on olfactory learning of a pollinator is interesting. Here we showed that bees were able to learn both odors (guava and mango), either via scented-US reward (Day 1 model) or via experience with the same odor each day (Day 2 model).
Thank you the question.
The statistical analyses are not described in enough detail to determine if the data were correctly analyzed. The authors seem to confuse linear regression with ANOVA at one point (line 219), and from my reading of the text, regressions were not actually used to test for learning. Perhaps, you used an analysis of covariance or multiple regression with day as a factor and learning as a continuous variable? The analyses need to be described in much more detail before an assessment of their appropriateness can be made. Based on my understanding of the statistical analyses, the figures do not reflect how the data were analyzed. The figures should reflect the analyses, illustrating what factors were determined to be significantly different and how. I recommend that the authors use a different approach to illustrating their data.We agree with this suggestion, and we have decided to analyze our data using the more robust approach of generalized linear mixed models. This change was appropriate because our response variable (bee olfactory learning performance, measured as the conditioned PER response to scent), can be modeled under a binomial distribution. Furthermore, the experimental design allows us to evaluate the effect of using scented unconditioned stimulus (US) in the CS-US pairing training, as well the effect of experience with same and different odor on day 2. Also, we included as fixed factor trials, given that training is a relevant aspect in learning. We also incorporate bee (individuals) as a random factor in our analyses.
Additional notes:
Line 103: explain why the bees were placed in darkness at room temperature before the test commenced.
For the experiments of absolute and differential conditioning we adapted the methods in Mc Cabe et al. 2007 and Abramson &Boyd 2001 with some modifications. Apparently, the rationale is to reduce possible stress related to storing them prior to the conditioning procedure [Lines 148-150].
“Each bee was fed 5 μl 35% (w/v) sucrose solution and then, to reduce potential stress, placed in darkness at room temperature for 3 hrs prior to an experiment.”
Line 106: what do you mean by “basic”? Do you mean to say that it is required for studies like this? Explain why that would be.
Thank you for the question. The evaluation of the Unconditioned Response (UR) PER response (i.e., the Proboscis Extension Reflex elicited when the antenna makes contact with sucrose solution) as well as the process of assessing Sugar Response Threshold (SRT), are first necessary steps when investigating if a bee species can be scent conditioned. For example, UR has not been reported in solitary bee species and therefore olfactory learning using PER is not applicable to those species. Although bees, in general prefer highly concentrated nectar in the range 30-50% sugar concentration, the information is to wide to be implemented in the conditioning protocol. The SRT has application on the ecological role of bees. For example, in the honey bee it has been shown that pollen foragers have a lower SRT than nectar foragers.
Now the paragraph in the Supplementary material 1 reads:
“Sucrose Response Threshold (SRT): Sugar Response Threshold (SRT) is the minimal sugar concentration at which the PER response occurs, and is basic in PER conditioning studies (Page et al. 1998). SRT is useful to know the range and the optimal sugar concentration to which species, colonies, or individuals respond. Here we evaluate SRT to determine the sucrose solution to which the stingless bee M. eburnea respond better and use that value in the conditioning procedure.”.
Line 114: Odd word arrangement. I would recommend “…its proboscis completely.” Corrected
Line 121: Provide a rational e for the use of 50% (w/v) sucrose solution, even if it is explained in greater detail after you describe the results of your SRT.
This value of sucrose concentration had been used in previous studies of STR with stingless bee and we wanted to keep as close as possible to the procedure follow in those studies. We specifically followed Mc Cabe et al. 2007):
Mc Cabe, S.I.; Hartfelder, K.; Santana, W.C.; Farina, W.M. Odor discrimination in classical conditioning of proboscis extension in two stingless bee species in comparison to Africanized honeybees. J. Comp. Physiol. A 2007, 193, 1089–1099. https://doi.org/10.1007/s00359-007-0260-8
Line 129: Even if your methods have been previously described, a brief description of them is still required for the new audience.
Corrected.
Line 150: replace “works” with “workers”.
Corrected.
Line 268: Reference where the MANOVA results are presented. The actual statistical parameters should be presented somewhere in the text.
This is resolved now by using generalized linear mixed models to analyze the data. Now we present the statistical parameters of the new analyses in Table2, Table 3, Table S1, Table S2 (Results and Supplementary material 2).
Thank you so much for yor time and effort. Our manuscript has improved thanks your and other Reviwers suggestions.
